# Sociodemographic profile, functionality, depression, and frailty as determinants for the risk of abuse and violence against older people in the community: An observational study conducted in Brazil

**Bruna Caroline Cassiano da Silva**[1]*, **Bruno Araújo da Silva Dantas**[2‡],
**Alexandre do Amaral Maculan**[3‡], **Nathaly da Luz Andrade**[1‡],
**Sandra Maria da Solidade Gomes Simões de Oliveira Torres**[1‡],
**Carmelo Sergio Gómez Martínez**[4‡], **Eulália Maria Chaves Maia**[1‡],
**Vilani Medeiros de Araújo Nunes**[5‡], **Maria Laurência Grou Parreirinha Gemito**[6‡],
**Sheila Cristina Rocha-Brischiliari**[3‡], **Gilson de Vasconcelos Torres**[1‡]

**1** Centro de Ciências da Saúde, Universidade Federal do Rio Grande do Norte (UFRN), Natal-RN, Brasil, **2** Faculdade de Ciências da Saúde do Trairi, Universidade Federal do Rio Grande do Norte (UFRN), Santa Cruz-RN, Brasil, **3** Universidade Estadual do Oeste do Paraná, Foz do Iguaçu-PR, Brasil, **4** Facultad de Enfermería, Universidad Catolica de Murcia (UCAM), Murcia, España, **5** Departamento de Saúde Coletiva, Universidade Federal do Rio Grande do Norte (UFRN), Natal-RN, Brasil, **6** Escola Superior de Enfermagem São João de Deus, Universidade de Évora, Évora, Portugal

☯ These authors contributed equally to this work.
‡ BADSD, ADAM, NDLA, SMDSGS, CSGM, EMCM and VMDAN, MLGPG, SCRB, GDVT also contributed equally to this work.
* brunacassiano1@gmail.com

## Abstract

This study aimed to analyze the relationship between sociodemographic factors, health, functionality, depression, and frailty with the risk of abuse and violence against younger and older adults individuals. A cross-sectional observational study with a quantitative approach was conducted among Brazilian older adults between April and July 2022. Participants aged 60 years and older were recruited from Brazilian Primary Health Care Units. The Hwalek-Sengstock Elder Abuse Screening Test, Lawton and Brody's Instrumental Activities of Daily Living Scale, Edmonton Frail Scale, and the Geriatric Depression Scale (GDS-15) were used to assess the variables of interest. Odds Ratios (ORs) was calculated using binary logistic regression models to test the study hypothesis. The sample was divided into two groups: younger elderly individuals (aged 60–70 years) and older elderly individuals (> 70 years). A total of n = 200 individuals' participants were included in the study (n = 132 younger and n = 68 older). Non-white skin color (n = 15/ 22.1%/ p = 0.016/ OR= 2.0) was identified as a risk factor for the older group, while illiteracy emerged as a risk factor for violence in both groups (OR> 1.0). The absence of depressive symptoms and frailty were protective factors against the risk of abuse and violence in both groups (OR>1.0). Logistic regression

**Data availability statement:** The research data was deposited on the Mendeley Data platform, accessible through the link: https://data.mende-ley.com/datasets/3kt8gyckfb.

**Funding:** This research was funded by the National Council for Scientific and Technological Development (CNPq), Brazil, Funding was obtained through CNPq/MCTI/FNDCT Call No. 18/2021 – Range B – Consolidated Groups, under grant number 0257801662000850; Coordenação de Aperfeiçoamento de Pessoal de Nível Superior (CAPES), Award Number: PRINT-PROGRAMA INSTITUCIONAL DE INTERNACIONALIZAÇÃO - 88887.831236/2023-00 - Edital n° 41/2017 and Fundação Norte-riograndense de Pesquisa e Cultura, Award Number: N°139/2021/PROPESQ/REITORIA/CONSUNI/UFRN. The funders had no role in study design, data collection and analysis, decision to publish, or preparation of the manuscript.

**Competing interests:** The authors have declared that no competing interests exist.

analysis indicated that depression was the variable most strongly associated with the risk of abuse and violence, particularly in the younger group (R²= 0.46/ p < 0.001/ ß = 0.56). Among the observed associations, non-white skin color was a risk factor for abuse and violence in the older group, whereas literacy, absence of depression, and absence of frailty were protective factors in both groups.

## Introduction

The ongoing inversion of the age pyramid, coupled with the rapid growth of the older population, has generated significant implications for society [1]. In Brazil, projections indicate that by 2060, the number of individuals in this age group will reach 73 million, equivalent to 32.7% of the country's population [2]. Recognizing that the global demographic aging brings a range of repercussions and challenges, abuse and violence against the older individuals stand out as a critical social issue worldwide [3].

Elder abuse and violence result from a multifactorial and complex process, widely recognized as public health priority [4,5]. This complexity was highlighted in a conceptual analysis of elder abuse, which identified factors such as female gender, low social support, and low-income conditions as key contributors. Elder abuse can manifest in various forms, including threats or intimidation, intentional use of physical harm, unauthorized resources use, forced sex acts, and punitive food restrictions [6]. These phenomena encompasses any action or inaction (either isolated or repeated) within an interpersonal relationship that causes harm or suffering to the individual, violating the trust between the victim and the perpetrator. The consequences can include severe physical injuries, psychological trauma, hospitalizations, or even death [7,8].

Prevalence rates of elder abuse vary widely across studies [9]. A systematic review and meta-analysis estimate a global prevalence of 15.7% [10], and with projected population growth, this figure could rise to nearly 19 million cases [11]. In Brazil, more than 30,000 elder abuse reports were registered in the first half of 2022 [12], representing an 87.0% increase by 2023 [13], further underscoring the growing importance of addressing this issue [14]. Psychological and financial abuse were the most frequently forms of elder abuse reported in 2019 across several Brazilian cities. Additionally, most perpetrators resided with the victims [15]. Another study reported a 35% prevalence of psychological violence and 4.5% for physical violence in a Brazilian community, with functional dependency being more common among older adults who experienced some form of violence [16].

International data also highlight older ages as a significant risk factor elder abuse. Victims aged 80 years or older are disproportionately affected, both globally and in Brazil [17]. Factors such as frailty, depression, and functional dependence further exacerbate the risk of abuse and violence [18–21]. Frailty is characterized by a decline in an physiological and functional reserves [22]. Geriatric depression is marked by profound sadness, often unrelated to external events, and is frequently linked to feelings of isolation and abandonment [23]. Functional dependence refers to the inability to perform self-care and daily activities that require intact physical and cognitive capacities [24].

Despite growing societal social awareness [25], obtaining accurate data on elder abuse remains challenging due to underreporting and definitional inconsistencies [5,26–28]. Reports of elder abuse represent only the visible portion of the problem, leaving a substantial number of unreported cases hidden beneath the surface [29]. Nonetheless, identifying and understanding the factors associated with elder abuse and violence is essential for planning targeted health interventions to prevent abuse, promote safety, and foster a culture of peace [4,14,25].

The rationale for this study lies in its focus on the factors associated with the risk of abuse and violence among older adults. By shedding light on these factors, the study aims to contribute to the development of effective prevention and intervention strategies, ultimately improving the quality of life, dignity, and well-being of older individuals, especially those most vulnerable.

The objective of this study was to analyze the relationship between sociodemographic aspects, health, functionality, depression, and frailty with the risk of abuse and violence against younger and older elderly individuals. We hypothesized that these factors are associated with the risk of abuse and violence to varying degrees in this population.

## Materials and methods

### Ethical aspects

This research was approved by the Ethics and Research Committee of the State University of Western Paraná, Brazil, (Opinion No. 4.662.293/2021). Before data collection, participants signed a written informed consent form indicating their agreement to participate. For illiterate participants, the consent form was read aloud, and upon agreement, their finger-print was collected as a signature. The research activities did not disrupt the operational hours of the Primary Health Care (PHC) units, as participants were approached either after their consultations or at a convenient time without interfering with healthcare services.

All ethical principles and recommendations for research involving human subjects were strictly followed in compliance with current Brazilian legislation and the Declaration of Helsinki, which outlines essential criteria for good scientific practices. Participants were informed about the purpose, risks, and benefits of the study, and confidentiality of the provided information was assured. No rewards or incentives were offered to either participants or interviewers.

### Study design and location

This observational, cross-sectional study with a quantitative approach was conducted in 2022 among older adults registered in PHC units in Foz do Iguaçu, Paraná, Brazil. The study is part of an international multicenter group titled "International Network for Research on Vulnerability, Health, Safety, and Quality of Life of the Older People: Brazil, Portugal, Spain, France, Chile, Mexico and the United States of America" in which the researchers are active members.

Foz do Iguaçu has 29 PHC units, organized into five health districts. None of the selected PHC units reported refusals to participate.

### Population and sample

The research population consisted of older people served by PHC in the study region, which an estimated population of 34,172 older adults according to official Brazilian data [30]. Based on this reference, the sample size was calculated using a 95% confidence level (Z = 1.96), a 5% sampling error (e = 0.05), an estimated accuracy proportion (P) of 50%, and an expected error (Q) of 50%. The resulting sample size was estimated to be 384 participants.

A non-probability convenience sampling was used. PHC staff provided a list of addresses and phone numbers of potential participants who had previously expressed interest in participating. Researchers contacted these individuals to schedule interviews based on mutual availability. The final sample included 200 participants who were available and accessible to the researchers during the designated data collection period. Consequently, the sample size was below the ideal representativeness.

Inclusion criteria were: (i) age ≥ 60 years (the threshold for older adults in Brazil); (ii) permanence resident in the region for at least six months; and (iii) registration in the Brazilian public health system. Exclusion criteria were: (i) cognitive impairment, defined as score <17 on the Mini-Mental State Examination [31]; and (ii) medical diagnosis of intellectual, neurological, or mental disabilities that could hinder motor or cognitive tasks required by the instruments.

## Data collection and availability

Data collection was conducted from April to July 2022 at PHC units through face-to-face interviews, involving only the researcher and the participant. Interviews were conducted in Portuguese, the native language of Brazil, and no translators were required. To ensure privacy, reserved spaces were provided by the PHC units, creating a safe and comfortable environment for participants.

The research team consisted of trained undergraduate and graduate nursing students. Training sessions, led by the research coordinators, ensured consistency in data collection. Interviews lasted approximately 40 minutes, and responses were recorded on the Google Forms platform (Alphabet Inc., California, USA) or printed questionnaires when internet access was unavailable. Neither participants nor interviewers were blinded.

## Instruments and variables

A structured sociodemographic and health questionnaire, developed and validated by ten PhD researchers in the field of gerontology linked to the research network of which the authors are part, was used for data collection. Before its implementation, pilot interviews were conducted with older adults who were not part of the sample to refine the instrument. Subsequently, the researchers jointly discussed necessary adjustments and produced the final version. The questionnaire encompassed the following variables: sex, age, educational attainment (categorized as literate or illiterate), and self-reported skin color (classified as white or non-white, with the latter including Black, mixed-race, Indigenous, or other racial identities). Health-related variables included the presence of self-reported diseases and the number of daily medications taken. Participants classified as literate were those capable of reading and writing, demonstrating a basic understanding of the alphabetic system. The sociodemographic and health variables were used to describe the profile of the study groups and to identify potential confounding factors.

The Hwalek-Sengstock Elder Abuse Screening Test (H-S/EAST) was utilized to assess the risk of elder abuse and violence. This instrument comprises dichotomous (yes/no) questions designed to identify conflicts between the individual and their family members or cohabitants. An example of a question includes whether the respondent is coerced into performing activities against their will. The total score is obtained by summing affirmative responses, with higher scores indicating an elevated risk of abuse and violence. The test has a scoring range of 0 to 15, with a threshold of ≥3 points indicating a risk of abuse and violence [32,33]. For the purposes of this study, the variable was categorized as "No risk" (<3 points) and "At risk" (≥3 points) and was adopted as the outcome variable.

To evaluate functional capacity, the Lawton and Brody Instrumental Activities of Daily Living (IADL) Scale [34,35] was administered. This instrument assesses an individual's ability to perform daily tasks, including mobility, verbal fluency, technological skills (e.g., phone use), food preparation, and medication management. The total score ranges from 7 to 21 points, with lower scores indicating reduced functional independence. While the original classification distinguishes between "Independent," "Partially Dependent," and "Totally Dependent" individuals [36], this study adopted a binary classification: "Dependent" (including both total and partial dependence) and "Independent", which was considered an exposure variable.

The Geriatric Depression Scale – 15-item version (GDS-15) was used to screen for depressive symptoms. This instrument consists of questions addressing emotional distress, including feelings of hopelessness, suicidal ideation, and excessive worry. Each question is answered with "yes" or "no," generating a total score ranging from 0 to 15 points. Based on this score, individuals are classified into the categories "No depression," "Mild depression," or "Severe depression" [23].

 

For analytical purposes, this variable was dichotomized into "With Depression" (including both mild and severe cases) and "Without Depression". It was adopted as an exposure variable.

Frailty status was evaluated using the Edmonton Frail Scale (EFS), which incorporates brief tasks administered by the interviewer, such as drawing clock hands to indicate a specified time. This instrument assesses multiple domains, including cognition, health status, functional capacity, and social support. Scores range from 0 to 17, with higher scores indicating greater frailty. The classification system defines individuals as "Not Frail" (0–4 points), "Pre-Frail" (5–6 points), and "Frail" (≥7 points) [22,37]. In this study, the variable was recategorized into "Frail" (including both "Pre-Frail" and "Frail") and "Not Frail", which was considered an exposure variable.

The risk of elder abuse and violence (H-S/EAST) was defined as the dependent variable, and the independent variables were gender, skin color, marital status, education, paid work, retirement, income, self-reported disease, number of medications, and the scores of functionality (Lawton & Brody), depression (GDS-15), and frailty (EFS). The sample was divided into two study groups according to age: one group of younger individuals aged between 60 and 70 years, and another of older individuals over 70 years. This division was made because the median age variable was 70 years, considered the cutoff point for age group categorization.

All instruments were translated and validated in the Brazilian Portuguese version, which was the language used in the questionnaires and during the interviews.

## Data analysis and processing

Data were organized and tabulated using Microsoft Excel version 2010 software (Microsoft Corporation, Washington, WA, USA), exported, and analyzed using the Statistical Package for the Social Sciences (SPSS) version 22.0 statistical software (IBM, Armonk, NY, USA). The Kolmogorov-Smirnov test indicated non-normal distribution of the data, and non-parametric tests were applied. Quantitative variables were categorized and presented as absolute frequencies (n) and relative frequencies (%).

The associations between the variable elder abuse risk (H-S/EAST) and sociodemographic profile, functionality (Lawton & Brody), frailty (EFS), and depressive symptoms (GDS-15) were tested using Pearson's chi-square test, a non-parametric test, to assess the independence between them.

Binary logistic regression models was used to measure the prediction of study group variables and variables identified as potential confounding factors based on the independent variables. The data were synthesized, highlighting $R^2$ (Nagelkerke), the Model LR value, Wald value, Beta (ß), standard error of Exp (ß) (S.E.), and Exp (ß) (OR). For all tests, the Odds Ratio (OR) was calculated, with an OR > 1.00 indicating a positive association between the exposure and the outcome. The collinearity test was performed to verify whether the independent variables met the criteria for binary logistic regression. To do so, linear regression was applied to the scalar independent variables. We adopted an Acceptable Tolerance >0.100 and Variance Inflation Factor (VIF) with the following parameters: VIF = 1.00 (no multicollinearity); VIF > 1.00 and < 5.00 (moderate multicollinearity); VIF > 5.00 and < 10.00 (high multicollinearity); VIF > 10.00 (Severe multicollinearity). A 5.0% margin of error and a 95% CI were adopted, with a significance level set at $p < 0.05$ for all analyses [38,39].

Reliability of the applied scales was assessed using Cronbach's Alpha, resulting in the following outcomes: H-S/EAST (crude α = 0.821/ adjusted α = 0.819), Lawton & Brody (crude α = 0.893/ adjusted α = 0.896), GDS-15 (crude α = 0.821/ adjusted α = 0.818), and EFS (crude α = 0.734/ adjusted α = 0.730), indicating substantial reliability. [40]. There was no missing data.

## Results

The study included 200 participants, divided into younger (n = 132) and older (n = 68) groups. Risk of elder abuse and violence (H-S/EAST) was identified in 87 participants (43.5%). The sample selection process is shown in Fig 1.

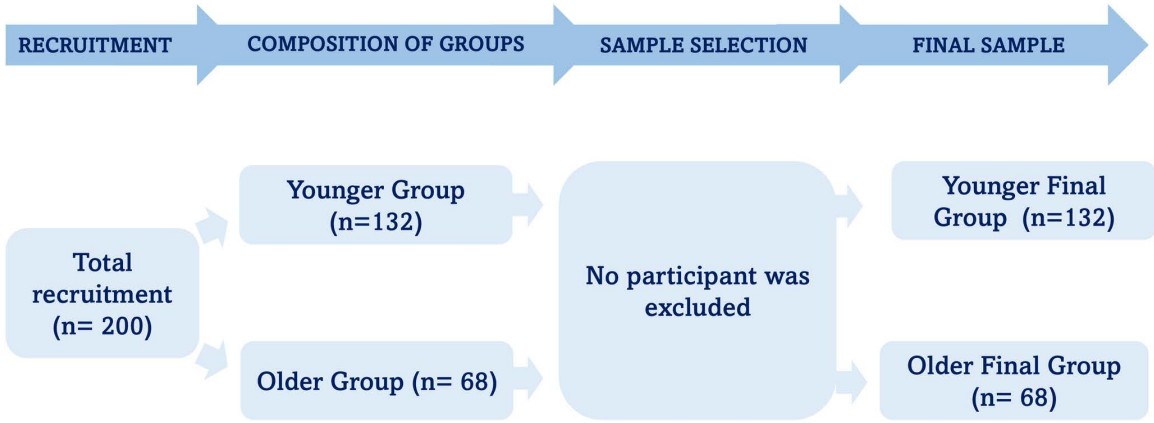

RECRUITMENT → COMPOSITION OF GROUPS → SAMPLE SELECTION → FINAL SAMPLE

Total recruitment (n= 200)

Younger Group (n=132)

Older Group (n= 68)

No participant was excluded

Younger Final Group (n=132)

Older Final Group (n= 68)

**Fig 1. Flowchart of the sample recruitment and selection process.**

The sample comprised predominantly woman (66.5%), white participants (58.5%), individuals with partners (62.5%), and literate individuals (78.0%). Most participants were unemployed or without regular income (83.5%), earned ≤2 minimum wages (75.5%), and were retired (65.5%). Most reported ≤2 self-reported diseases (74.0%) and used ≤2 daily medications (53.5%).

Table 1 illustrates the association between the risk of abuse and violence (H-S/EAST) sociodemographic and health variables in the two study groups. In the older group, being of non-white race emerged as a risk factor for violence (H-S/EAST) (n = 15/ 22.1%/ p = 0.016/ OR= 2.0). Literacy was identified as a protective factor in both the younger group (n = 66/ 50.0%/ p = 0.015/ OR= 2.0) and the older group (n = 33/ 48.5%/ p = 0.002/ OR= 2.3).

When evaluating functionality (Lawton & Brody), depressive symptoms (GDS-15), and frailty (EFS) in the total sample (n = 200), we observed a predominance of functionally dependent participants (60.0%), individuals without depression (71.0%), and those classified as not frail (69.0%).

Table 2 presents the analysis of association between the risk of abuse and violence (H-S/EAST) and these variables according to the age groups. Among younger individuals, functional independence was found to be a protective factor against the risk of abuse and violence (n = 45/ 34.1%/ p = 0.001/ OR= 1.7). Conversely, in the older group, functional dependence was associated with a higher likelihood of absence of risk of abuse and violence (n = 27/ 39.7%/ p = 0.037/ OR= 1.6). In other analyses, the absence of depressive symptoms (GDS-15) and the absence of frailty (EFS) were protective factors against the risk of abuse and violence across both groups, with ORs greater than 1.0.

Logistic regression analysis was conducted on the variables with positive OR from the previous tests, as summarized in Table 3. Among younger individuals, depressive symptoms (GDS-15) were identified as the strongest predictor of risk for abuse and violence (R²= 0.46/ p < 0.001/ ß = 0.56/ OR= 1.7). Similarly, in the older group, depressive symptoms remained a key predictor, albeit with slightly lower predictive values (R²= 0.33/ p < 0.001/ ß = 0.43/ OR= 1.5). The variables considered as potential confounding factors were tested against the independent variables. No substantial adjustments were found regarding Functionality, Depressive Symptoms or Frailty. Despite some variations in R² and OR, it was not possible to establish that Skin Color and Education influenced the independent variables. A more detailed analysis is available in S1 Table.

Additionally, we conducted an analysis of the coefficient of collinearity between the independent variables, all of which exhibited an acceptable tolerance for binary logistic regression (>0.1) and a moderate level of multicollinearity (VIF > 1.00 and < 5.00). A detailed breakdown of this analysis is available in S2 Table. The number of interactions obtained between the variables ranged from 4 to 5 and can be accessed in S3 Table.

**Table 1. Odds Ratios of the risk of abuse and violence (H-S/EAST) and sociodemographic and health variables, with reference to study groups.**

| Sociodemographic and health characterization | Risk of abuse and violence (H-S/EAST) (n = 200) | | | | | | | |
|---|---|---|---|---|---|---|---|---|
| | Younger (n = 132) | | | | Older (n = 68) | | | |
| | No Risk (n = 73) | At risk (n = 59) | p[a] | OR (CI95%) | No Risk (n = 40) | At risk (n = 28) | p[a] | OR (CI95%) |
| | | n (%) | n (%) | | | n (%) | n (%) | |
| **Gender** | | | | | | | | |
| Woman | 51 (38.6) | 39 (29.5) | 0.645 | 1.2 (0.6–2.5) | 24 (35.3) | 19 (27.9) | 0.645 | 0.7 (0.3–2.0) |
| Men | 22 (16.7) | 20 (15.2) | | | 16 (23.5) | 9 (13.2) | | |
| **Skin color** | | | | | | | | |
| No white[b] | 27 (20.5) | 31 (23.5) | 0.073 | 0.5 (0.3–1.1) | 10 (14.7) | 15 (22.1) | 0.016 | 2.0 (1.3–3.5) |
| White | 46 (34.8) | 28 (21.2) | | | 30 (44.1) | 13 (19.1) | | |
| **Marital status** | | | | | | | | |
| Single/widowed/divorced | 20 (15.2) | 22 (16.7) | 0.225 | 0.6 (0.3–1.3) | 16 (23.5) | 17 (25.0) | 0.139 | 0.4 (0.2–1.2) |
| Married/stable union | 53 (40.2) | 37 (28.0) | | | 24 (35.3) | 11 (16.2) | | |
| **Education** | | | | | | | | |
| Literate[c] | 66 (50.0) | 44 (33.3) | 0.015 | 2.0 (1.0–3.5) | 33 (48,5) | 13 (19.1) | 0.002 | 2.3 (1.2–4.3) |
| Illiterate | 7 (5.3) | 15 (11.4) | | | 7 (10,3) | 15 (22.1) | | |
| **Paid Work** | | | | | | | | |
| No | 58 (43.9) | 44 (33.3) | 0.506 | 1.3 (0.6–3.0) | 39 (57.4) | 26 (38.2) | 0.359 | 3.0 (0.3–34.8) |
| Yes | 15 (11.4) | 15 (11.4) | | | 1 (1.5) | 2 (2.9) | | |
| **Retired** | | | | | | | | |
| No | 33 (25.0) | 25 (18.9) | 0.744 | 1.1 (0.6–2.2) | 5 (7.4) | 6 (8.8) | 0.325 | 0.5 (1.4–1.9) |
| Yes | 40 (30.3) | 34 (25.8) | | | 35 (51.5) | 22 (32.4) | | |
| **Income** | | | | | | | | |
| 0-2 minimum wage[d] | 51 (38.6) | 40 (30.3) | 0.799 | 1.1 (0.5–2.3) | 36 (52.9) | 24 (35.3) | 0.589 | 1.5 (0.3–6.6) |
| > 2 minimum wage | 22 (16.7) | 19 (14.4) | | | 4 (5.9) | 4 (5.9) | | |
| **Number of Self-Reported Diseases** | | | | | | | | |
| 0-2 | 55 (41.7) | 42 (31.8) | 0.591 | 1.2 (0.6–2.7) | 30 (44.1) | 21 (30.9) | 1.000 | 1.0 (0.3–3.0) |
| ≥ 3 | 18 (13.6) | 17 (12.9) | | | 10 (14.7) | 7 (10.3) | | |
| **Number of Daily Medications** | | | | | | | | |
| 0-2 | 32 (24.2) | 32 (24.2) | 0.234 | 0.7 (0.3–1.3) | 27 (39.7) | 16 (23.5) | 0.383 | 1.6 (0.6–4.2) |
| ≥ 3 | 41 (31.1) | 27 (20.5) | | | 13 (19.1) | 12 (17.6) | | |

[a]Pearson's chi-squared test;

[b]Individuals who identified as black, mixed-race, indigenous, or other non-white categories.

[c]who can read and write, demonstrating recognition of the alphabetic system.

[d]Minimum wage in Brazil in 2022 = 1,212.00 BRL (Approximately 231.00 USD in the same year).

Younger: 60–70 years.

Older: > 70 years.

## Discussion

The main findings of this study revealed that non-white skin color was a risk factor exclusively among older individuals (>70 years). Higher education levels (literate status), the absence of depressive symptoms (GDS-15), and the absence of frailty (EFS) emerged as significant protective factors against the risk of abuse and violence in both groups. Functional independence (Lawton & Brody) was a protective factor exclusively among younger individuals. These findings address

**Table 2. Odds Ratios of the risk of abuse and violence (H-S/EAST), functionality, depressive symptoms, and frailty, with reference to age groups.**

| Aspects evaluated | Risk of abuse and violence (H-S/EAST) (n = 200) | | | | | | | |
| --- | --- | --- | --- | --- | --- | --- | --- | --- |
| | Younger (n = 132) | | | | Older (n = 68) | | | |
| | No Risk (n = 73) | At risk (n = 59) | pª | OR (CI95%) | No Risk (n = 40) | At risk (n = 28) | pª | OR (CI95%) |
| | | n (%) | n (%) | | | n (%) | n (%) | |
| **Functionality (Lawton & Brody)** | | | | | | | | |
| Independent | 45 (34.1) | 19 (14.4) | 0.001 | 1.7 (1.2–2.4) | 13 (19.1) | 3 (4.4) | 0.037 | 1.6 (1.1–2.2) |
| Dependent | 28 (21.2) | 40 (30.3) | | | 27 (39.7) | 25 (36.8) | | |
| **Depressive Symptoms (GDS-15)** | | | | | | | | |
| Without depression | 68 (51.5) | 29 (22.0) | <0.001 | 4.9 (2.2–11.2) | 35 (51.5) | 10 (14.7) | <0.001 | 3.6 (1,6–7.9) |
| With depression | 5 (3.8) | 30 (22.7) | | | 5 (7.4) | 18 (26.5) | | |
| **Frailty (EFS)** | | | | | | | | |
| Not frail | 61 (46.2) | 35 (26.5) | 0.002 | 2.0 (1.8–3.1) | 30 (44.1) | 12 (17.6) | 0.007 | 2.0 (1.1–3.1) |
| Frail | 12 (9.1) | 24 (18.2) | | | 10 (14.7) | 16 (23.5) | | |

ªPearson's chi-squared test;

ᵇOR (Odds Ratio);

Younger: 60–70 years.

Older: >70 years.

critical gaps in the scientific literature by clarifying how factors associated with abuse and violence differ according to age strata within the older population.

Although some studies do not include skin color as a variable [14,16,17,41] or or fail to identify its association with violence against older adults [42,43], in Brazil, skin color partially reflects the social and health conditions of older individuals [44]. In this study, non-white individuals in the older group exhibited a higher risk of abuse and violence. Scholars have argued that racial discrimination and social exclusion exacerbate vulnerabilities, increasing susceptibility to abuse and violence across all age groups [45]. Furthermore, non-white older adults in Brazil experience pronounced face social and health inequalities, often occupying the lowest socioeconomic and health status strata [44]. Conversely, other studies shown that victims of all forms of violence are predominantly white [42,46,47], suggesting that the relationship between race and abuse may vary across contexts and methodologies.

Literacy was a significant protective factor against abuse and violence in both age groups, aligning with findings from other studies [17,25,41]. Education is known to enhance access to information and healthcare services, empowering individuals to recognize and report abuse. Scholars suggest that education fosters autonomy and decision-making skills, thereby reducing vulnerability to abuse, which can lead to severe physical injuries and functional impairments [16,48]. Despite playing an important role in the outcome of violence [49], some studies [17,41] found no significant influence, while others suggest that older people with higher levels of education may feel embarrassed to report abuse, choosing not to seek support [50]. Thus, while literacy can mitigate vulnerability, higher education alone does not universally serve as a protective factor.

Functional independence, as assessed by the Lawton & Brody IADL scale, was identified as a protective factor against abuse and violence among younger individuals. However, dependence appeared to reduce the risk of abuse and violence in the older group, a seemingly contradictory result. Generally, independence in daily activities reduces exposure to abusive situations. Supporting this, a Chinese study reported that neglect by caregivers, often hired to assist individuals with physical or functional dependence, was the most frequent form of violence experienced by older adults [51]. Other studies corroborate the association between functional dependence and higher rates of violence [52], particularly among

**Table 3. Logistic regression analysis of the most significant aspects and variables, adjusted for potential confounding factors according to age groups.**

| Risk of abuse and violence (HS-EAST) | | Younger (n = 132) | | | | | Older (n = 68) | | | | |
|---|---|---|---|---|---|---|---|---|---|---|---|
| | | R²ª | pᵇ | ß (S.E)ᶜ | Wald | OR (CI 95%) | R² | pᵇ | ß (S.E)ᶜ | Wald | OR (CI 95%) |
| **Aspects evaluated** | | | | | | | | | | | |
| **Functionality (Lawton & Brody) – Scalar** | | 0.17 | <0.001 | −0.31 (0.10) | 12.10 | 0.7 (0.6–0.9) | 0.35 | <0.001 | −0.30 (0.10) | 14.20 | 0.7 (0.6–0.9) |
| Adjusted by Skin color | No whiteᵈ | 0.17 | 0.023 | −0.35 (0.15) | 5.19 | 0.7 (0.5–0.9) | 0.49 | 0.037 | −0.53 (0.26) | 4.37 | 0.6 (0.3–1.0) |
| | White | 0.16 | 0.010 | 0.28 (0.11) | 6.68 | 0.7 (0.6–0.9) | 0.29 | 0.005 | −0.27 (0.09) | 8.04 | 0.8 (0.6–0.9) |
| Adjusted by Education | Literateᵉ | 0.12 | 0.005 | −0.28 (0.10) | 7.73 | 0.8 (0.6–0.9) | 0.24 | 0.007 | −0.26 (0.10) | 7.23 | 0.8 (0.6–0.9) |
| | Illiterate | - ᶠ | – | – | – | – | 0.31 | 0.058 | −0.30 (0.16) | 3.60 | 0.7 (0.5–1.0) |
| **Depressive Symptoms (GDS-15) – Scalar** | | 0.46 | <0.001 | 0.56 (0.10) | 32.65 | 1.7 (1.4–2.1) | 0.33 | <0.001 | 0.43 (0.12) | 13.60 | 1.5 (1.2–2.0) |
| Adjusted by Skin color | No whiteᵈ | 0.43 | <0.001 | 0.51 (0.13) | 14.39 | 1.6 (1.6–2.2) | 0.40 | 0.015 | 0.46 (0.19) | 5.97 | 1.6 (1.1–2.3) |
| | White | 0.45 | <0.001 | 0.59 (0.14) | 17.20 | 1.8 (1.3–2.4) | 0.32 | 0.007 | 0.47 (0.17) | 7.23 | 1.6 (1.1–2.3) |
| Adjusted by Education | Literateᵉ | 0.42 | <0.001 | 0.54 (0.11) | 25.37 | 1.7 (1.4–2.1) | 0.30 | 0.005 | 0.42 (0.15) | 7.80 | 1.5 (1.1–2.0) |
| | Illiterate | 0.54 | 0.027 | 0.69 (0.31) | 4.89 | 2.0 (1.1–3.7) | 0.31 | 0.043 | 0.45 (0.22) | 4.09 | 1.6 (1.0–2.4) |
| **Frailty (EFS) – Scalar** | | 0.19 | <0.001 | 0.27 (0.07) | 16.44 | 1.3 (1.2–1.5) | 0.24 | <0.001 | 0.27 (0.08) | 10.67 | 1.3 (1.1–1.6) |
| Adjusted by Skin color | No whiteᵈ | 0.15 | 0.019 | 0.25 (0.11) | 5.46 | 1.3 (1.0–1.6) | 0.44 | 0.012 | 0.43 (0.17) | 6.26 | 1.5 (1.1–2.1) |
| | White | 0.20 | 0.002 | 0.27 (0.01) | 9.81 | 1.3 (1.1–1.5) | 0.16 | 0.033 | 0.23 (0.11) | 4.54 | 1.3 (1.0–1.6) |
| Adjusted by Education | Literateᵉ | 0.15 | 0.001 | 0.25 (0.07) | 10.97 | 1.3 (1.1–1.5) | 0.18 | 0.020 | 0.26 (0.11) | 5.37 | 1.3 (1.0–1.6) |
| | Illiterate | – | – | – | – | – | – | – | – | – | – |

ªR² de Nagelkerke.

ᵇModel (Forward LR).

ᶜUnstandardized coefficient.

ᵈIndividuals who identified as black, mixed-race, indigenous, or other non-white categories.

ᵉwho can read and write, demonstrating recognition of the alphabetic system.

ᶠValues were not calculated by the statistical program because the variable did not meet the criteria for regression analysis.

Younger: 60-70 years.

Older: >70 years.

individuals aged 60–70 years [25]. However, our logistic regression analysis did not attribute significant predictive value to functionality in determining abuse risk. Similarly, some studies have found no association between functional dependence and abuse [17,53], highlighting that not all victims are frail or dependent [54]. While functional capacity influences autonomy, it does not appear to be a decisive determinant of abuse risk.

Regarding depressive symptoms (GDS-15), the absence of depression served as a protective factor in both age groups, with logistic regression analysis identifying depression as the most predictive variable for abuse and violence. Other studies have similarly noted a strong association between depression and violence among older adults [14,17,26,27,47]. Depressed older individuals often report feelings of loneliness, isolation, and a lack of social support [27,54], which may increase their vulnerability. Emotional states such as hopelessness, fear, and low self-esteem can deter individuals from reporting abuse [55]. A previous study found that depression among older adults was associated with increased emotional and physical abuse by caregivers and family members, who exploit the victim's fragile emotional state to perpetrate financial abuse and establish dominance [20].

Frailty (EFS) was another critical variable in this study, It was identified as a significant risk factor for violence in both age groups, both in categorical and logistic regression analyses, similar to other findings in the literature [4,26,47,56]. Frailty, characterized by a decrease in physiological reserves and physical resistance, increases the dependence of older people for daily activities, making them more susceptible to abuse [56]. Marked by a decrease in willingness for activities

and the ability to cope with stress, frailty syndrome is becoming more common in the older population, significantly contributing to various types of violence [14]. Although the study conducted by Brijoux, Neise (17), found no significant influence between frailty and the risk of violence, other research highlights that older people at risk of abuse and violence are 4.24 times more likely to be frail [56], and individuals with frailty syndrome are 3.07 times more likely to suffer abuse [14].

Overall, this study demonstrates that sociodemographic, health, functionality, depression, and frailty aspects can influence the risk of abuse and violence to varying extents, depending on the individual's age group. Violence against older adults is a pressing global issue with severe implications for their health and well-being. Given its status as a public health and social problem, there is a critical need for proactive measures to identify risk factors and potential abuse situations early. Addressing vulnerabilities in older populations can support both preventive interventions and protective strategies for those at risk.

## Limitation of study

This study has several limitations. The cross-sectional design precludes the determination of causal relationships between variables, limiting the ability to establish whether the identified factors are causes or consequences of abuse risk. The smaller sample size of older participants (>70 years) may have constrained the power of intergroup comparisons. Furthermore, the study was conducted in a single municipality, which may limit the generalizability of its findings to other socially and culturally distinct regions. Skin Color and Education behaved as potential confounding variables in the crude analysis with the study groups (Age groups). However, we conducted a more detailed analysis to assess whether they could influence the independent variables (Functionality, Depressive Symptoms and Frailty), and this hypothesis was ruled out. Nevertheless, we acknowledge that the small sample size may have influenced the adjustment analyses.

Data collection coincided with the resumption of healthcare services following COVID-19 restrictions, potentially reducing access to more frail or older individuals who continued to avoid healthcare facilities. Additionally, reliance on self-reported data may have introduced response biases, particularly on sensitive topics such as violence. Participants may have underreported or omitted abuse experiences due to fear of retaliation or social stigma. To mitigate these biases, interviews were conducted in private settings, ensuring confidentiality.

## Conclusion

Our study identified associations between sociodemographic factors, functionality, depression, frailty, and the risk of abuse and violence among older adults. Non-white race emerged as a risk factor among older participants. Literacy, absence of depression, and absence of frailty were the primary protective factors across both age groups, while preserved functionality was protective exclusively among younger individuals. These findings support our study hypothesis.

These findings underscore the importance of considering multiple factors when assessing the risk of abuse and violence in older populations. Interventions aimed at improving education, enhancing mental health, and strengthening functionality and independence can effectively reduce abuse risk. Public policies should prioritize protecting vulnerable subgroups, particularly racial minorities and individuals with lower educational attainment.

Further research is warranted, including clinical trials and intervention studies, to deepen understanding and develop strategies to address violence against older adults effectively.

## Supporting information

**S1 Table. Detailed logistic regression analysis of the most significant aspects and variables, adjusted for potential confounding factors according to age groups.**
(DOCX)

**S2 Table. Collinearity coefficient analysis between the independent variables according to age group.**
(DOCX)

 

**S3 Table. Analysis of the interaction between the independent variable for each age group and total sample.**
(DOCX)

## Acknowledge

We especially thank the participants of our study, the managers of the Primary Health Care units who assisted in the recruitment and selection process of the sample and allowed us access to the physical space and available information in the service. We also extend our gratitude to the undergraduate students who assisted in data collection.

## Author contributions

**Conceptualization:** Bruna Caroline Cassiano da Silva, Bruno Araújo da Silva Dantas, Nathaly da Luz Andrade, Maria Laurência Grou Parreirinha Gemito, Sheila Cristina Rocha-Brischiliari, Gilson de Vasconcelos Torres.

**Data curation:** Nathaly da Luz Andrade, Carmelo Sergio Gómez Martínez, Sheila Cristina Rocha-Brischiliari, Gilson de Vasconcelos Torres.

**Formal analysis:** Bruno Araújo da Silva Dantas, Gilson de Vasconcelos Torres.

**Funding acquisition:** Gilson de Vasconcelos Torres.

**Investigation:** Bruna Caroline Cassiano da Silva, Alexandre do Amaral Maculan, Maria Laurência Grou Parreirinha Gemito, Sheila Cristina Rocha-Brischiliari.

**Methodology:** Bruna Caroline Cassiano da Silva, Bruno Araújo da Silva Dantas, Alexandre do Amaral Maculan, Nathaly da Luz Andrade, Sandra Maria da Solidade Gomes Simões de Oliveira Torres, Eulália Maria Chaves Maia, Vilani Medeiros de Araújo Nunes, Maria Laurência Grou Parreirinha Gemito, Sheila Cristina Rocha-Brischiliari, Gilson de Vasconcelos Torres.

**Project administration:** Gilson de Vasconcelos Torres.

**Resources:** Gilson de Vasconcelos Torres.

**Supervision:** Bruno Araújo da Silva Dantas, Carmelo Sergio Gómez Martínez, Eulália Maria Chaves Maia, Vilani Medeiros de Araújo Nunes, Maria Laurência Grou Parreirinha Gemito, Sheila Cristina Rocha-Brischiliari, Gilson de Vasconcelos Torres.

**Validation:** Carmelo Sergio Gómez Martínez, Eulália Maria Chaves Maia, Sheila Cristina Rocha-Brischiliari, Gilson de Vasconcelos Torres.

**Visualization:** Bruno Araújo da Silva Dantas, Sandra Maria da Solidade Gomes Simões de Oliveira Torres, Eulália Maria Chaves Maia, Vilani Medeiros de Araújo Nunes, Maria Laurência Grou Parreirinha Gemito, Sheila Cristina Rocha-Brischiliari, Gilson de Vasconcelos Torres.

**Writing – original draft:** Bruna Caroline Cassiano da Silva, Nathaly da Luz Andrade, Sandra Maria da Solidade Gomes Simões de Oliveira Torres.

**Writing – review & editing:** Bruno Araújo da Silva Dantas, Eulália Maria Chaves Maia, Vilani Medeiros de Araújo Nunes, Sheila Cristina Rocha-Brischiliari, Gilson de Vasconcelos Torres.

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
