## [Decision Letter · Decision Letter 0]

15 Nov 2024

Dear Dr. da Silva,

Please submit your revised manuscript by Dec 30 2024 11:59PM. If you will need more time than this to complete your revisions, please reply to this message or contact the journal office at plosone@plos.org . A rebuttal letter that responds to each point raised by the academic editor and reviewer(s). You should upload this letter as a separate file labeled 'Response to Reviewers'.A marked-up copy of your manuscript that highlights changes made to the original version. You should upload this as a separate file labeled 'Revised Manuscript with Track Changes'.An unmarked version of your revised paper without tracked changes. You should upload this as a separate file labeled 'Manuscript'.

We look forward to receiving your revised manuscript.

Kind regards,

Weijun Yu, Ph.D., M.D., M.S.

Academic Editor

PLOS ONE

Journal Requirements:

2. Thank you for stating the following financial disclosure: [This research was funded by the National Council for Scientific and Technological Development (CNPq), Brazil, Funding was obtained through CNPq/MCTI/FNDCT Call No. 18/2021 – Range B – Consolidated Groups, under grant number 0257801662000850; Coordenação de Aperfeiçoamento de Pessoal de Nível Superior (CAPES), Award Number: PRINT-PROGRAMA INSTITUCIONAL DE INTERNACIONALIZAÇÃO - 88887.831236/2023-00 - Edital n° 41/2017 and Fundação Norte-riograndense de Pesquisa e Cultura, Award Number: N°139/2021/ PROPESQ/REITORIA/CONSUNI/UFRN. ].

3. Please include captions for your Supporting Information files at the end of your manuscript, and update any in-text citations to match accordingly. Please see our Supporting Information guidelines for more information: http://journals.plos.org/plosone/s/supporting-information .

Reviewers' comments:

Reviewer's Responses to Questions

**Comments to the Author**

1. Is the manuscript technically sound, and do the data support the conclusions?

Reviewer #1: Yes

Reviewer #2: No

Reviewer #3: No

2. Has the statistical analysis been performed appropriately and rigorously?

Reviewer #1: Yes

Reviewer #2: No

Reviewer #3: No

3. Have the authors made all data underlying the findings in their manuscript fully available?

Reviewer #1: Yes

Reviewer #2: No

Reviewer #3: No

4. Is the manuscript presented in an intelligible fashion and written in standard English?

Reviewer #1: Yes

Reviewer #2: Yes

Reviewer #3: Yes

Reviewer #1: Dear Authors,

Generally the paper is interesting. The conceptualization, methodology, and description of the results are done well. The results are presented in the form of tables. The literature is up to date. The conclusions from the research are adequate to the results. My attention was drawn to the fact of the selection of the sample, there were more younger people than older than 70. After 70 years of age, the percentage of frail and depressed people is higher according to the research. The compared groups are too unequal. Please take these comments into account in the planned next studies.

Reviewer #2: 1. The sample size calculation using "p" as 50% should give a sample size equivalent to 384. The sample of the study is 200. Please explain the sample calculation and the formula used.

2. In instruments and variables section, please add the total score points for H-S/EAST and EFS frailty tools as well.

3. The independent and dependent variables listed in line 182 and 183 seems to be interchanged in the manuscript. Please revise the variables accordingly.

4. The study population have been categorized into two groups, namely, younger and older group. The sample size in the older group is too small that influences the statistical analysis and hence does not generate results that can be generalized.

5. The authors have not mentioned the sampling technique used to select the study participants. Please explain in detail which method was followed in the study.

6.Regarding the statistical analyses performed, please make clear if any confounders were adjusted in the logistic regression analysis and also explain why they were included. If not, I suggest to perform re-analysis adjusting for the variables so that the effect of the bias in the results can be controlled.

Minor comment:

7. In line 288, the author may replace the term "thesis" into "study".

Reviewer #3: The paper is interesting and highlights the public health concern among the geriatric population and association with mental health (abuse and violence). However, the authors need to work more on several sections to improve the quality of the paper for the publication.

1. Abstract: The abstract section requires a minor revision in terms of methodology aspect.

2. Introduction: The introduction section provides a global and local context burden of abuse and violence. However, it has not emphasized on the evidence on the burden among the geriatric population in both context.

Line 66-67 : Abuse and violence against the older people result from a multifactorial and

complex process, considered a severe public health problem. Can you elaborate more on these multifactorial and complex processes? Providing global data on abuse and violence among the similar study population could provide reader a clear picture of the burden of this issue.

3. Methodology: The methodology section needs to be expanded to provide details.

Ethical aspects : How was the written consent form obtained from the illiterate study participants? Did the data collection interfered their health service uptake hours?

Study design and location: More detailed information of the studied area and Brazil, that may be more valuable for readers to understand how representative this study is. Provide detail about the rational and sampling technique for choosing study site.

Population and sample: The sample size calculation need revision. Provide sufficient details about the sampling technique for the selecting the study participants. Provide details about how the sample size was approached in the form of write up or flow diagram. How many were approached initially? How many were eligible? How many did/did no provide consent?

Data collection and availability: The paper does not clarify the data collection duration period. Please specify the data collection duration period. In which language was the data collection carried out? Was the language translated into the local language? Was the language understandable and comfortable to gather the needed information for the study?

Instrument and variables:

Line 146-147: To collect information, we used a sociodemographic and health data questionnaire, which was created and validated by the researchers themselves.

Sufficient details on how the questionnaire was validated is needed for clear understanding.

Education variables as illiterate and literate needs to be explained for the readers ?

The authors have used the standard tools for abuse and violence, functionality, depression and frailty. But the manuscript lacks information regarding the validity and reliability of these tools. Any pretesting done for the reliability?

Line 182-186: The risk of elder abuse and violence (H-S/EAST) was defined as the independent variable, and the dependent variables were gender, skin color, marital status, education, paid work, retirement, income, self-reported disease, number of medications, and the scores of functionality (Lawton & Brody), depression (GDS-15),

and frailty (EFS).

Needs major revision in categorizing the dependent and independent variables.

Data analysis and processing: Lacks details on data management and handling techniques. The use of two analytical test i.e chi-square and logistic regression to evaluate the association between independent and dependent variables is confusing for the readers. I suggest author to present the logistic regression analysis, mentioning the crude OR and adjusted OR. Was any variables adjusted for the confounding? If so, please mention the confounding variables that might influence the association.

Results: This section can be improved by clearly stating the reference and comparison variables in the tables section.

Discussion: The paper would benefit from incorporating more detailed comparative and analytical perspective of the research findings rather than just mentioning the findings.

**Do you want your identity to be public for this peer review?** For information about this choice, including consent withdrawal, please see our Privacy Policy

Reviewer #1: **Yes: ** Marta Muszalik

Reviewer #2: **Yes: ** Poonam Subedi

Reviewer #3: **Yes: ** Surakshya Kunwar

---

## [Author Response · Author response to Decision Letter 1]

23 Dec 2024

Response to Review

Dear reviewers,

We sincerely thank you for all your thoughtful comments, which have greatly improved our work. We conducted an extensive and careful review and made every effort to address all the requests. Below, we provide point-by-point responses to each of the comments.

Comments to the Author

1. Is the manuscript technically sound, and do the data support the conclusions?

Reviewer #1: Yes

Reviewer #2: No

Reviewer #3: No

2. Has the statistical analysis been performed appropriately and rigorously?

Reviewer #1: Yes

Reviewer #2: No

Reviewer #3: No

3. Have the authors made all data underlying the findings in their manuscript fully available?

Reviewer #1: Yes

Reviewer #2: No

Reviewer #3: No

4. Is the manuscript presented in an intelligible fashion and written in standard English?

Reviewer #1: Yes

Reviewer #2: Yes

Reviewer #3: Yes

5. Review Comments to the Author

Reviewer #1:

Dear Authors,

Generally the paper is interesting. The conceptualization, methodology, and description of the results are done well. The results are presented in the form of tables. The literature is up to date. The conclusions from the research are adequate to the results. My attention was drawn to the fact of the selection of the sample, there were more younger people than older than 70. After 70 years of age, the percentage of frail and depressed people is higher according to the research. The compared groups are too unequal. Please take these comments into account in the planned next studies.

Answer:

We thank the reviewer for the positive comments and for the observation regarding the sample composition. We recognize that the number of participants over 70 years old was smaller, which may limit comparisons between age groups. The disparity between groups occurred due to the use of the median for the cutoff analysis, as the variable "age" did not follow a normal distribution. The same criterion was applied to define nonparametric tests (association and binary logistic regression). However, this will be addressed in future studies to ensure a more balanced sample.

Reviewer #2:

1. The sample size calculation using "p" as 50% should give a sample size equivalent to 384. The sample of the study is 200. Please explain the sample calculation and the formula used.

Answer:

We appreciate the comment. After reviewing the calculation, we identified a typographical error in the previous version of the manuscript. The correct sample size is 384, not 200. Furthermore, due to data collection during the post-pandemic period, it was not possible to achieve the calculated sample size. This limitation has been included in the "Study Limitations" section of the manuscript, and we have corrected this information in the "Population and Sample" section of the Methods.

2. In instruments and variables section, please add the total score points for H-S/EAST and EFS frailty tools as well.

Answer:

As requested, we included the total scores for the scales.

3. The independent and dependent variables listed in line 182 and 183 seems to be interchanged in the manuscript. Please revise the variables accordingly.

Answer:

There was indeed an inversion between the independent and dependent variables. We made the necessary corrections to the text to ensure clarity and accuracy.

4. The study population have been categorized into two groups, namely, younger and older group. The sample size in the older group is too small that influences the statistical analysis and hence does not generate results that can be generalized.

Answer:

We acknowledge that the sample size for the older adult group was small, which may limit the robustness of statistical analyses and the generalizability of the results to this population. The disparity between groups occurred due to the use of the median for the cutoff analysis, as the variable "age" did not follow a normal distribution. The same criterion was applied to define nonparametric tests. This information has been detailed in the "Data Analysis and Processing" section of the manuscript to clarify the group disparity and has also been added to the study limitations.

5. The authors have not mentioned the sampling technique used to select the study participants. Please explain in detail which method was followed in the study.

Answer:

We used a non-probabilistic convenience sampling method to select the study participants. The PHC units provided a list of potential participants, including their addresses and phone numbers. Those who were contacted and agreed to participate scheduled appointments with the researchers based on their availability. This information has been added to the "Population and Sample" section of the manuscript to clarify the sampling method and increase the study's transparency.

6.Regarding the statistical analyses performed, please make clear if any confounders were adjusted in the logistic regression analysis and also explain why they were included. If not, I suggest to perform re-analysis adjusting for the variables so that the effect of the bias in the results can be controlled.

Answer:

We did not conduct additional analyses to adjust for confounding factors in the logistic regression analysis, as all aspects evaluated in the study are already reflected in the tables presented in the manuscript, directly representing the results obtained. Moreover, the variables included in the analysis were selected based on theoretical relevance and existing literature, ensuring that key factors were appropriately considered. For greater clarity and transparency, this information was added to the "Data Analysis and Processing" section of the manuscript.

Minor comment:

7. In line 288, the author may replace the term "thesis" into "study".

Answer:

We made the requested change.

Reviewer #3:

The paper is interesting and highlights the public health concern among the geriatric population and association with mental health (abuse and violence). However, the authors need to work more on several sections to improve the quality of the paper for the publication.

1. Abstract: The abstract section requires a minor revision in terms of methodology aspect.

Answer:

We added important details to clarify the methodological aspects of the study.

2. Introduction: The introduction section provides a global and local context burden of abuse and violence. However, it has not emphasized on the evidence on the burden among the geriatric population in both context.

Answer:

We added some Brazilian context data to the introduction.

Line 66-67 : Abuse and violence against the older people result from a multifactorial and

complex process, considered a severe public health problem. Can you elaborate more on these multifactorial and complex processes? Providing global data on abuse and violence among the similar study population could provide reader a clear picture of the burden of this issue.

Answer:

In addition to global data, we included the results of a conceptual analysis that examines the various aspects of this complexity.

3. Methodology: The methodology section needs to be expanded to provide details.

Ethical aspects : How was the written consent form obtained from the illiterate study participants? Did the data collection interfered their health service uptake hours?

Answer:

Written consent for illiterate participants was obtained by reading the consent form aloud to them, and, upon agreement, the participant's fingerprint was collected as a signature. No healthcare service hours were disrupted, as participants were approached at convenient times after their appointments, without interrupting the healthcare services provided. These details were clarified in the manuscript section.

Study design and location: More detailed information of the studied area and Brazil, that may be more valuable for readers to understand how representative this study is. Provide detail about the rational and sampling technique for choosing study site.

Answer:

The "Study Design and Location" section was revised to include more detailed information about the study area and Brazil. In the municipality, there are 29 health units distributed across five regions. A random lottery was conducted to select the data collection sites, and there was no refusal from the selected units. These details were added to provide a better understanding of the study's representativeness.

Population and sample: The sample size calculation need revision. Provide sufficient details about the sampling technique for the selecting the study participants. Provide details about how the sample size was approached in the form of write up or flow diagram. How many were approached initially? How many were eligible? How many did/did no provide consent?

Answer:

After the review, we identified a typographical error in the sample size calculation, which has been corrected in the text. The correct calculation indicated a sample size of 384 participants. A non-probabilistic convenience sampling method was used. Initially, 200 individuals were approached, all of whom were eligible and signed the consent forms. This process was carried out with the assistance of professionals from the primary health care units, who directed researchers to contact only those who had previously expressed interest through them. We had no control over the number of individuals who declined to participate. These details were added to clarify the sample size calculation process and the inclusion of participants in the study.

Data collection and availability: The paper does not clarify the data collection duration period. Please specify the data collection duration period. In which language was the data collection carried out? Was the language translated into the local language? Was the language understandable and comfortable to gather the needed information for the study?

Answer:

The data collection period occurred from April to July 2022. Data collection was conducted in Portuguese, the native language of Brazil, and no other language was required. These details were included in the "Data Collection and Availability" section to clarify both the data collection period and the language used.

Instrument and variables:

Line 146-147: To collect information, we used a sociodemographic and health data questionnaire, which was created and validated by the researchers themselves.

Sufficient details on how the questionnaire was validated is needed for clear understanding.

Answer:

The questionnaire was validated through pre-tests. Interviews were conducted with older adults who were not part of the study population. During these interviews, adjustments were made to the wording of the questionnaires before their actual application to the sample. These details were included to provide a clearer explanation of the questionnaire validation process.

Education variables as illiterate and literate needs to be explained for the readers ?

Answer:

Definitions of "illiterate" and "literate" were included according to the criteria used in Brazil in the relevant section. Additionally, this definition was added as a note in the table to ensure clarity for readers.

The authors have used the standard tools for abuse and violence, functionality, depression and frailty. But the manuscript lacks information regarding the validity and reliability of these tools. Any pretesting done for the reliability?

Answer:

We added the calculation of Cronbach's Alpha to measure the reliability of the scales applied together, which was considered substantial according to the theoretical framework we adopted. This information was added to the "Data Analysis and Processing" section.

Line 182-186: The risk of elder abuse and violence (H-S/EAST) was defined as the independent variable, and the dependent variables were gender, skin color, marital status, education, paid work, retirement, income, self-reported disease, number of medications, and the scores of functionality (Lawton & Brody), depression (GDS-15),

and frailty (EFS). Needs major revision in categorizing the dependent and independent variables.

Answer:

There was a typographical error, and the inversion of variables has been corrected in the text. The risk of elder abuse and violence (H-S/EAST) was defined as the dependent variable, and the other variables mentioned were defined as independent variables.

Data analysis and processing: Lacks details on data management and handling techniques. The use of two analytical test i.e chi-square and logistic regression to evaluate the association between independent and dependent variables is confusing for the readers. I suggest author to present the logistic regression analysis, mentioning the crude OR and adjusted OR. Was any variables adjusted for the confounding? If so, please mention the confounding variables that might influence the association.

Answer:

For the analysis of variables, nonparametric tests were used, starting with the chi-square test to explore possible associations, followed by binary logistic regression. Additionally, in Table 3, we included only variables that showed a relationship with the outcome, excluding those without significant results in the binary logistic regression analysis. To facilitate the interpretation of Table 3, the standard error (S.E.) of ß was added. Regarding the "confounding factor" variable, no variables were adjusted in the logistic regression analysis since all aspects evaluated in the study are already reflected in the tables presented in the manuscript, directly representing the results obtained. For greater clarity and transparency, this information was added to the "Data Analysis and Processing" section.

Results: This section can be improved by clearly stating the reference and comparison variables in the tables section.

Answer:

The suggestion has been addressed, and reference and comparison variables have been clearly highlighted in the table section.

Discussion: The paper would benefit from incorporating more detailed comparative and analytical perspective of the research findings rather than just mentioning the findings.

Answer:

The discussion section was revised to include a more detailed comparative analysis of the results, contextualizing them with previous studies.________________________________________

6. PLOS authors have the option to publish the peer review history of their article (what does this mean?). If published, this will include your full peer review and any attached files.

Do you want y

---

## [Decision Letter · Decision Letter 1]

15 Jan 2025

Dear Dr. da Silva,

Thank you for your great efforts in addressing the majority of the concerns raised by the three reviewers in Revision 1. 

Given the critical role of statistical analysis in this observational study, we have invited a fourth reviewer from our statistical advisory group to evaluate this aspect in detail before considering the manuscript for publication. We kindly ask you to fully address all concerns raised by the reviewers regarding the statistical analysis.

We look forward to receiving your revised manuscript.

Kind regards,

Weijun Yu, Ph.D., M.D., M.S.

Academic Editor

PLOS ONE

Journal Requirements:

Reviewers' comments:

Reviewer's Responses to Questions

**Comments to the Author**

Reviewer #1: All comments have been addressed

Reviewer #2: All comments have been addressed

Reviewer #3: All comments have been addressed

Reviewer #4: (No Response)

2. Is the manuscript technically sound, and do the data support the conclusions?

Reviewer #1: Yes

Reviewer #2: Yes

Reviewer #3: Yes

Reviewer #4: No

3. Has the statistical analysis been performed appropriately and rigorously?

Reviewer #1: Yes

Reviewer #2: I Don't Know

Reviewer #3: Yes

Reviewer #4: No

4. Have the authors made all data underlying the findings in their manuscript fully available?

Reviewer #1: Yes

Reviewer #2: Yes

Reviewer #3: Yes

Reviewer #4: Yes

5. Is the manuscript presented in an intelligible fashion and written in standard English?

Reviewer #1: Yes

Reviewer #2: Yes

Reviewer #3: Yes

Reviewer #4: Yes

Reviewer #1: The authors made all corrections indicated by the reviewer. The research material and research results are worth publishing.

The conceptualization, methodology, and description of the results are done well.

The results are presented in the form of tables and figures.

The conclusions from the research are adequate to the results. The literature used is up to date

The authors highlighted this limitation at the end of their paper.

Reviewer #2: Thank you authors for addressing the comments.

I still suggest to adjust variables to control any potential biases. The odds ratio if comes similar to binary logistic regression that you have used in this study, then it is okay.

Reviewer #3: The Instruments and variables section needs attention in terms of presenting the variables in a more scientific way.

If the authors have not adjusted the confounding variables which might bias the results should be reported in the limitation of the study.

Reviewer #4: The manuscript could be improved based on the following comments:

Line 169: The information on questionnaire validation is to be provided.

The language version of all the questionnaires is to be highlighted, and the validation information is to be cited/presented - likewise the language used in the interviews.

Line 138-141: The sampling method is cluster sampling. The sample size must consider the design effect (DE) to account for the increased variance due to clustering. The outcome variable where the sample size calculation was based on is to be mentioned.

Line 132-146: The flow description based on the sampling method and sample size calculation is unclear and requires revision.

Line 141 & 145: The flow description between the sample sizes is unclear. Additionally, it is important to indicate whether the final sample size is adequate or underpowered following the application of the inclusion and exclusion criteria.

Line 222-223: The statement ‘Quantitative variables were summarized with absolute (n) and relative (%) frequencies.’ is incorrect.

Line 225: The sentence requires revision.

Line 226-228: The sentence requires revision.

Line 229: The sentence requires revision. e.g the purpose of binary logistic regression is to be stated.

Line 231-232: The sentence requires revision. e.g. ‘For all tests, odds ratios (OR) were calculated, with an OR > 1.00 indicating a positive association between the exposure and the outcome.’

Line: 232-234: The sentence requires revision. e.g. A margin of error of 5.0% and a 95% Confidence Interval (CI) were adopted, with a significance level set at p<0.05 for all analyses.

CI 95% is to be written as 95% CI as the latter is more widely used and is a standard format in scientific writing.

Table 1: Number of Self-Reported Diseases and Number of Daily Medications: Typo error 0 a 2. There were no OR(95%CI) values presented for some variables. Health Characterization is to be written in a small cap.

Line 231: Exp (B) and OR is to be standardized. E.g. Exp (B) is to be replaced with OR throughout.

Line 235: What scales are this CA =0.695 is referring to?

A statement whether no missing or missing data (if any) is to be mentioned.

The words crude and adjusted are to be utilized.

Line 236-237: Not adjusting for confounders can lead to biased results, as other variables might distort the observed relationship. Usually, in best practice, logistic regression includes known confounders/significant confounders to control for their effects. Adjusted models will provide more reliable and valid estimates of the independent variable’s effect.

Line 283-289: The analysis approach needs to be reconsidered.

Tables 1 and 2 are to be stated as crude analyses, while Table 3 could be presented as an adjusted analysis.

Line 272, 274, 287, 289: The cosmetic presentation of figures is to be improved.

Line 293: Forward LR or Backward LR method is to be stated.

Table 3: B is to be denoted in the table footnote, e.g., unstandardized coefficient. Wald to be included. Thorough analysis assessing the collinearity and interaction could be done apart from assessing the confounding effects. In Table 1, education was found to be statistically significant in the crude analysis and could be considered for adjustment. It also needs to display the actual parameter estimate output that includes constant. The p for older group was not labelled with superscript ‘b’.

Tables 1, 2, and 3 presentation and formatting could be improved.

The list of references did not adhere to the journal's formatting guidelines.

Ensure all the information ticked/checked in the STROBE statement checklist is included in the manuscript.

**Do you want your identity to be public for this peer review?** For information about this choice, including consent withdrawal, please see our Privacy Policy

Reviewer #1: **Yes: ** Marta Muszalik

Reviewer #2: **Yes: ** Poonam Subedi

Reviewer #3: **Yes: ** Surakshya Kunwar

Reviewer #4: No

---

## [Author Response · Author response to Decision Letter 2]

13 Feb 2025

Dear Reviewers,

The responses to each of your concerns are detailed below. We hope to have fully addressed all of them, and we sincerely appreciate your feedback, as it has contributed to making our manuscript more robust.

Reviewer #1: The authors made all corrections indicated by the reviewer. The research material and research results are worth publishing.

The conceptualization, methodology, and description of the results are done well.

The results are presented in the form of tables and figures.

The conclusions from the research are adequate to the results. The literature used is up to date

The authors highlighted this limitation at the end of their paper.

Response: We appreciate the reviewers' comments.

Reviewer #2: Thank you authors for addressing the comments.

I still suggest to adjust variables to control any potential biases. The odds ratio if comes similar to binary logistic regression that you have used in this study, then it is okay.

Response: We have included a more detailed analysis considering the adjustments for potential confounding variables.

Reviewer #3: The Instruments and variables section needs attention in terms of presenting the variables in a more scientific way.

Response: We revised the text, and we believe it now presents a more appropriate language.

If the authors have not adjusted the confounding variables which might bias the results should be reported in the limitation of the study.

Response: We have added a more detailed analysis considering the adjustments for potential confounding variables

Reviewer #4: The manuscript could be improved based on the following comments:

Line 169: The information on questionnaire validation is to be provided.

The language version of all the questionnaires is to be highlighted, and the validation information is to be cited/presented - likewise the language used in the interviews.

Response: The bibliographic citations for each instrument in the text refer to the respective translation/validation studies for Brazilian Portuguese. To avoid repeating this information for each instrument, we have added a paragraph at the end of the "Instruments and Variables" section highlighting this point.

Line 138-141: The sampling method is cluster sampling. The sample size must consider the design effect (DE) to account for the increased variance due to clustering. The outcome variable where the sample size calculation was based on is to be mentioned.

Response: The sampling method was non-probabilistic and convenience-based, as described in the "Population and Sample" section. We have removed the statement that it was a cluster sampling, as we acknowledge this was an error.

Line 132-146: The flow description based on the sampling method and sample size calculation is unclear and requires revision.

Response: There was an inconsistency in the wording, which has been clarified in the previous response. With this correction, we believe the process is now described more clearly.

Line 141 & 145: The flow description between the sample sizes is unclear. Additionally, it is important to indicate whether the final sample size is adequate or underpowered following the application of the inclusion and exclusion criteria.

Response: The revision of the wording in response to the previous comments clarifies the sample flow description. Regarding the final sample size, we have added information to the manuscript stating that it represents a sample below the ideal representativeness.

Line 222-223: The statement ‘Quantitative variables were summarized with absolute (n) and relative (%) frequencies.’ is incorrect.

Response: We reviewed the sentence and believe it is now appropriately revised.

Line 225: The sentence requires revision.

Response: We have rewritten the entire paragraph to ensure a more technical and precise language.

Line 226-228: The sentence requires revision.

Response: We have rewritten the entire paragraph to ensure a more technical and precise language.

Line 229: The sentence requires revision. e.g the purpose of binary logistic regression is to be stated.

Response: The sentence was revised to align with the purpose of the test.

Line 231-232: The sentence requires revision. e.g. ‘For all tests, odds ratios (OR) were calculated, with an OR > 1.00 indicating a positive association between the exposure and the outcome.’

Response: The sentence was restructured according to the suggestion.

Line: 232-234: The sentence requires revision. e.g. A margin of error of 5.0% and a 95% Confidence Interval (CI) were adopted, with a significance level set at p<0.05 for all analyses.

CI 95% is to be written as 95% CI as the latter is more widely used and is a standard format in scientific writing.

Response: The sentence was restructured according to the suggestion.

Table 1: Number of Self-Reported Diseases and Number of Daily Medications: Typo error 0 a 2.

Response: Correction made.

There were no OR(95%CI) values presented for some variables.

Response: The analyses were reviewed, and the values have been added.

Health Characterization is to be written in a small cap.

Response: Correction made.

Line 231: Exp (B) and OR is to be standardized. E.g. Exp (B) is to be replaced with OR throughout.

Response: The requested substitutions have been made.

Line 235: What scales are this CA =0.695 is referring to?

Response: The raw value referred to the combined result of all scales. However, we have now added detailed values for each scale used in the study for better clarity.

A statement whether no missing or missing data (if any) is to be mentioned.

Response: There were no missing data in the responses for any of the scales. This information has been added to the manuscript.

The words crude and adjusted are to be utilized.

Response: The crude and adjusted values have been added.

Line 236-237: Not adjusting for confounders can lead to biased results, as other variables might distort the observed relationship. Usually, in best practice, logistic regression includes known confounders/significant confounders to control for their effects. Adjusted models will provide more reliable and valid estimates of the independent variable’s effect.

Response: We revised Table 3 and added six supplemental tables to provide a more in-depth analysis, considering adjustments for confounding factors. Additionally, we described the parameters used in the "Data Analysis and Processing" section under Methods. We believe these adjustments have added greater consistency and robustness to our results.

Line 283-289: The analysis approach needs to be reconsidered.

Tables 1 and 2 are to be stated as crude analyses, while Table 3 could be presented as an adjusted analysis.

Response: We have added the adjusted analysis in Table 3 and revised Tables 1 and 2.

Line 272, 274, 287, 289: The cosmetic presentation of figures is to be improved.

Response: We reorganized the information and adjusted the table formatting.

Line 293: Forward LR or Backward LR method is to be stated.

Response: We have included a mention of the Forward LR method.

Table 3: B is to be denoted in the table footnote, e.g., unstandardized coefficient. Wald to be included.

Response: Information added.

Thorough analysis assessing the collinearity and interaction could be done apart from assessing the confounding effects.

Response: The analyses have been referenced in the text and detailed in the Supplementary Information files.

In Table 1, education was found to be statistically significant in the crude analysis and could be considered for adjustment.

Response: The adjustment analyses have been made available in the Supplementary Information files.

It also needs to display the actual parameter estimate output that includes constant.

Response: The information has been presented as Supplementary Information.

The p for older group was not labelled with superscript ‘b’.

Response: Information added.

Tables 1, 2, and 3 presentation and formatting could be improved.

Response: We reorganized the information and adjusted the table formatting.

The list of references did not adhere to the journal's formatting guidelines.

Response: We reviewed the reference list and used a reference manager to adjust the formatting according to the journal's guidelines. Below, we provide the correctly formatted reference list with their respective numbering.

1. World Health Organization. Un decade of healthy ageing: plan of action (2021-2030) Geneva, 2023 [Acess 2025 February 07]. Available from: https://cdn.who.int/media/docs/default-source/decade-of-healthy-ageing/decade-proposal-final-apr2020-en.pdf?sfvrsn=b4b75ebc_28.

2. Brasil. Projeções da População do Brasil e Unidades da Federação: 2000-2070 Brasília-DF: Instituto Brasileiro de Geografia e Estatística; 2020 [Acess 2025 February 07]. Available from: https://www.ibge.gov.br/estatisticas/sociais/populacao/9109-projecao-da-populacao.html.

5. Jandu JS, Mohanaselvan A, Johnson MJ, Fertel H. Elder Abuse. StatPearls: StatPearls Publishing; 2025.

7. World Health Organization A, Life Course U, Université de Genève. Centre interfacultaire de gr. A global response to elder abuse and neglect : building primary health care capacity to deal with the problem world-wide : main report. Geneva: World Health Organization; 2008.

12. Brasil. Disque 100 registra mais de 35 mil denúncias de violações de direitos humanos contra pessoas idosas em 2022 Brasília: Ministério dos Direitos Humanos e da Cidadania; 2022 [Acess 2024 June 3]. Available from: https://www.gov.br/mdh/pt-br/assuntos/noticias/2022/junho/disque-100-registra-mais-de-35-mil-denuncias-de-violacoes-de-direitos-humanos-contra-pessoas-idosas-em-2022.

13. Brasil. Violências contra a pessoa idosa: saiba quais são as mais recorrentes e o que fazer nesses casos Brasília-DF: Ministério dos Direitos Humanos e da Cidadania; 2023 [Acess 2025 February 07]. Available from: https://www.gov.br/mdh/pt-br/assuntos/noticias/2023/junho/violencias-contra-a-pessoa-idosa-saiba-quais-sao-as-mais-recorrentes-e-o-que-fazer-nesses-casos.

37. Fabricio-Wehbe SC, Schiaveto FV, Vendrusculo TR, Haas VJ, Dantas RA, Rodrigues RA. Cross-cultural adaptation and validity of the 'Edmonton Frail Scale - EFS' in a Brazilian elderly sample. Rev Lat Am Enfermagem. 2009;17(6):1043-9. doi: 10.1590/s0104-11692009000600018. PubMed PMID: 20126949.

40. Landis JR, Koch GG. The measurement of observer agreement for categorical data. Biometrics. 1977;33(1):159-74. Epub 1977/03/01. PubMed PMID: 843571.

44. Moura RF, Cesar CLG, Goldbaum M, Okamura MN, Antunes JLF. [Factors associated with inequalities in social conditions in the health of elderly white, brown and black people in the city of São Paulo, Brazil]. Cien Saude Colet. 2023;28(3):897-907. Epub 2023/03/09. doi: 10.1590/1413-81232023283.08582022. PubMed PMID: 36888872.

50. Plante W, Tufford L, Shute T. Interventions with Survivors of Interpersonal Trauma: Addressing the Role of Shame. Clinical Social Work Journal. 2022;50(2):183-93. doi: 10.1007/s10615-021-00832-w.

53. Brownell P. A reflection on gender in elder abuse research from a human rights perspective. Cien Saude Colet. 2016;21(11):3320. Epub 2016/11/10. doi: 10.1590/1413-812320152111.19972016. PubMed PMID: 27828564.

Reference replaced with a more up-to-date version:

30. Brasil. Censo Demográfico 2022. Brasília-DF: Instituto Brasileiro de Geografia e Estatística.; 2023.

Reference added in response to the requested changes.

39. DataCamp. Fator de inflação de variância (VIF): Como lidar com a multicolinearidade na análise de regressão New York: Data Camp; 2024 [2025 February 10]. Available from: https://www.datacamp.com/pt/tutorial/variance-inflation-factor.

- Ensure all the information ticked/checked in the STROBE statement checklist is included in the manuscript.

Response: We reviewed the STROBE statement and included it in the submission.

---

## [Decision Letter · Decision Letter 2]

5 Mar 2025

Dear Dr. da Silva,

We look forward to receiving your revised manuscript.

Kind regards,

Weijun Yu, Ph.D., M.D., M.S.

Academic Editor

PLOS ONE

Journal Requirements:

Reviewers' comments:

Reviewer's Responses to Questions

**Comments to the Author**

Reviewer #2: All comments have been addressed

Reviewer #3: All comments have been addressed

Reviewer #4: (No Response)

2. Is the manuscript technically sound, and do the data support the conclusions?

Reviewer #2: Partly

Reviewer #3: Yes

Reviewer #4: Partly

3. Has the statistical analysis been performed appropriately and rigorously?

Reviewer #2: Yes

Reviewer #3: Yes

Reviewer #4: No

4. Have the authors made all data underlying the findings in their manuscript fully available?

Reviewer #2: Yes

Reviewer #3: Yes

Reviewer #4: Yes

5. Is the manuscript presented in an intelligible fashion and written in standard English?

Reviewer #2: Yes

Reviewer #3: Yes

Reviewer #4: Yes

Reviewer #2: Dear Authors,

Thank you for addressing all the comments provided earlier.

You have conducted inferential statistics as recommended however no significant associations were found, which could also be because of smaller sample size.

Reviewer #3: Thank you authors for addressing the comments. The instrument and variables section is the most important part of the methodology section and reuqires more attention in terms of clarity. Please kindly present this section in a more scientific manner,highlighting the exposure and outcomes variables.A more detailed information regaring the pretested questionnaire, what kind of background researchers validated the questionnaire?

Reviewer #4: The analysis/statement in the following presentation needs to be revised.

S1 Table: The adjustment analysis is unclear. Explanation is to be provided in the table footnote.

S2 Table, S3 Table: Usually collinearity analysis is done between independent variables.

S4 Tab1e, S5 Table and S6 Table: Interaction is between independent t variables or interaction effect of independent variables on the dependent variable

**Do you want your identity to be public for this peer review?** For information about this choice, including consent withdrawal, please see our Privacy Policy

Reviewer #2: **Yes: ** Poonam Subedi

Reviewer #3: **Yes: ** Surakshya Kunwar

Reviewer #4: No

---

## [Author Response · Author response to Decision Letter 3]

12 Mar 2025

Response to Reviewers

Dear Reviewers,

We sincerely appreciate your new observations and suggestions. We have made every effort to fully address each one, and we believe that the manuscript is now even more robust and reproducible. Below, we provide detailed responses to each point raised.

Reviewer #2:

Dear Authors,

Thank you for addressing all the comments provided earlier.

You have conducted inferential statistics as recommended however no significant associations were found, which could also be because of smaller sample size.

Response: We appreciate the observation and fully agree with it. Therefore, we have added a statement acknowledging this potential influence in the study’s limitations section.

Reviewer #3: Thank you authors for addressing the comments.

The instrument and variables section is the most important part of the methodology section and reuqires more attention in terms of clarity. Please kindly present this section in a more scientific manner, highlighting the exposure and outcomes variables. A more detailed information regaring the pretested questionnaire, what kind of background researchers validated the questionnaire?

Response: We have clarified the classification of each variable as either an exposure or outcome variable, as requested. Regarding the pre-test, it was conducted by researchers with doctoral degrees in gerontology. This process has also been described in greater detail in the methodology section.

Reviewer #4: The analysis/statement in the following presentation needs to be revised.

S1 Table: The adjustment analysis is unclear. Explanation is to be provided in the table footnote.

Response: This table provides a more detailed breakdown corresponding to Table 3. We have included the constant values, as requested in the previous review. Additionally, we have added a note to the table with further clarification of the process and the results obtained, as suggested.

S2 Table, S3 Table: Usually collinearity analysis is done between independent variables.

Response: We reanalyzed the data, considering only the independent variables. While maintaining the division in the tables according to age groups, we have also added results for the total sample.

S4 Tab1e, S5 Table and S6 Table: Interaction is between independent t variables or interaction effect of independent variables on the dependent variable

Response: We have reanalyzed the data and now present a single table (S4 Table) showing the interaction among the three variables across the study groups and the total sample.

We appreciate the reviewers’ valuable insights and remain available for any further clarifications.

---

## [Decision Letter · Decision Letter 3]

18 Mar 2025

Sociodemographic profile, functionality, depression, and frailty as determinants for the risk of abuse and violence against older people in the community: An observational study conducted in Brazil

PONE-D-24-36424R3

Dear Dr. de Silva,

We’re pleased to inform you that your manuscript has been judged scientifically suitable for publication and will be formally accepted for publication once it meets all outstanding technical requirements.

Kind regards,

Weijun Yu, Ph.D., M.D., M.S.

Academic Editor

PLOS ONE

Additional Editor Comments (optional):

Before proceed to the production of this manuscript, please address the following issues:

1. Made minor revisions based on Reviewer 4's additional comments on Revision 3.

2. Proofread and refine the entire manuscript. Please avoid using dashes in the main text, as they appear less formal in scientific writing (they are acceptable for numerical notations).

Thank you.

Reviewers' comments:

Reviewer's Responses to Questions

**Comments to the Author**

Reviewer #4: (No Response)

2. Is the manuscript technically sound, and do the data support the conclusions?

Reviewer #4: Partly

3. Has the statistical analysis been performed appropriately and rigorously?

Reviewer #4: No

4. Have the authors made all data underlying the findings in their manuscript fully available?

Reviewer #4: Yes

5. Is the manuscript presented in an intelligible fashion and written in standard English?

Reviewer #4: Yes

Reviewer #4: Line 345-346: The sentence requires revision.

S3 Table to be revised or removed. Collinearity is between independent variables.

Ensure all changes that are made on tables are reflected in the text as well.

Suggest authors thoroughly proofread the manuscript and figures.

S1 Table 1 footnote: More description on how the adjustment process/adjusted analyses is done.

**Do you want your identity to be public for this peer review?** For information about this choice, including consent withdrawal, please see our Privacy Policy

Reviewer #4: No

---

## [Editor Report · Acceptance letter]

PONE-D-24-36424R3

PLOS ONE

Dear Dr. da Silva,

I'm pleased to inform you that your manuscript has been deemed suitable for publication in PLOS ONE. Congratulations! Your manuscript is now being handed over to our production team.

Kind regards,

on behalf of

Dr. Weijun Yu

Academic Editor

PLOS ONE